# GRADUAL LEARNING: OPTIMIZING FINE-TUNING WITH PARTIALLY MASTERED KNOWLEDGE IN LARGE LANGUAGE MODELS

## ABSTRACT

During the pretraining phase, large language models (LLMs) acquire vast amounts of knowledge from extensive text corpora. Nevertheless, in later stages such as fine-tuning and inference, the model may encounter knowledge not covered in the initial training, which can lead to hallucinations and degraded performance. This issue has a profound impact on the model's capabilities, as it will inevitably face out-of-scope knowledge after pretraining. Furthermore, fine-tuning is often required to adapt LLMs to domain-specific tasks, necessitating the acquisition of new knowledge. However, this phenomenon limits the model's ability to learn and integrate new information during fine-tuning. The effectiveness of fine-tuning largely depends on the type of knowledge involved. Existing research suggests that fine-tuning the model on partially mastered knowledge—for instance, question-answer pairs where the model has a chance of providing correct responses under non-greedy decoding—can enable the model to acquire new knowledge while mitigating the forgetting of previously learned information. Notably, this approach can still lead to the forgetting of fully mastered knowledge, constraining the fine-tuning dataset to a narrower range and limiting the model's overall potential for improvement. Given the model's intrinsic reasoning abilities and the interconnectedness of different knowledge areas, it is likely that as the model's capacity to utilize existing knowledge improves during fine-tuning, previously unmastered knowledge may become more understandable. To explore this hypothesis, we conducted experiments and, based on the results, proposed a two-stage fine-tuning strategy. This approach not only improves the model's overall test accuracy and knowledge retention but also preserves its accuracy on previously mastered content. When fine-tuning on the WikiQA dataset, our method increases the amount of knowledge acquired by the model in this stage by **24%**.

## 1 INTRODUCTION

In recent years, large language models (LLMs) have advanced rapidly, demonstrating exceptional performance across a variety of tasksDevlin et al. (2019); Achiam et al. (2023); Dubey et al. (2024); Yang et al. (2024). These models are pretrained on vast amounts of text data, allowing them to encode substantial knowledge within their parametersPetroni et al. (2019); Roberts et al. (2020); AlKhamissi et al. (2022). Following pretraining, LLMs undergo supervised fine-tuning to improve their ability to follow instructionsZhao et al. (2023); Zhang et al. (2023) and tackle domain-specific tasksHe et al. (2023); Cui et al. (2023); Li et al. (2023).

However, unlike the pretraining phase, LLMs face challenges when encountering new knowledge during other stages, including fine-tuning and inference. When fine-tuning data includes knowledge unrelated to the data seen during the pretraining phase, LLMs struggle to learn this new information, making them prone to overfitting and generating hallucinations.Gekhman et al. (2024); Gudibande et al. (2023). Additionally, even when external knowledge is integrated into prompts during inference, such as through KG-enhanced LLMsPan et al. (2024) or Retrieval-Augmented Generation (RAG)Zhao et al. (2023), these models often struggle to manage unfamiliar knowledge effectivelyKang et al. (2024); Lee et al. (2023); Yin et al. (2023). These characteristics impose the following constraints on the design of LLMs:

- **Constraint 1:** During the fine-tuning stage, the knowledge in the data should be associated with the knowledge from the pretraining phase to mitigate overfitting and reduce hallucinations.

- **Constraint 2:** During the inference stage, if external knowledge is to be provided in prompts, it must be related to the knowledge the model has already learned.

In this data-driven eraBai et al. (2024), these constraints fundamentally limit the data utilization during fine-tuning and inference, thereby significantly restricting the capabilities of LLMs. To mitigate the impact of these constraints on model performance, we propose a novel two-stage fine-tuning method based on the interconnections between knowledge and the model's reasoning ability. This approach aims to broaden the pool of data suitable for training during the fine-tuning stage.

In comparison to smaller models, LLMs exhibit surprising emergent abilities, such as reasoningWei et al. (2022). For instance, we can harness the knowledge reasoning capability of LLMs by utilizing KG to enhance the inputs of these models, thereby improving their performancePan et al. (2024). Additionally, the knowledge within these models is not isolated but rather interconnectedJi et al. (2021). Based on these observations, we hypothesize that if an LLM acquires knowledge that was not fully mastered—knowledge the model is likely to answer correctly without fine-tuning, during self-supervised fine-tuning, it could potentially leverage its existing knowledge to comprehend additional concepts that were neither previously mastered nor present in the training dataset. To test this hypothesis, we conducted experiments using mainstream LLMs on a closed-book question-answering task and validated our conjecture. Building on these findings, we designed a two-stage fine-tuning framework: in the first stage, the model is fine-tuned using knowledge that it partially understands; in the second stage, we monitor changes in the types of knowledge within the training set and use the data with improved mastery as augmented training data for a second stage of fine-tuning. The experimental results illustrate that our approach substantially improves the model's accuracy on test data and its mastery of knowledge within the training dataset.

Our contributions can be summarized as follows:

- We propose that fine-tuning can enhance a LLM's mastery of knowledge that it previously did not fully comprehend and that was not present in the original training dataset. We conducted experiments to validate this hypothesis.

- Building on this, we utilized the data with improved mastery to augment the training dataset and performed a subsequent round of fine-tuning on the LLM, which can significantly enhance knowledge acquisition and lead to increased accuracy on the test set. This method broadens the range of data available for fine-tuning LLMs, offering a new perspective from a data-driven standpoint to enhance the performance.

## 2 BACKGROUND AND RELATED WORK

### 2.1 IMPACT OF NEW KNOWLEDGE ON HALLUCINATION

LLMs can acquire substantial knowledge during the pretraining phase, enabling them to function as knowledge bases for downstream tasksAlKhamissi et al. (2022). However, these models are prone to a phenomenon known as hallucination, where the generated content, while syntactically correct and seemingly coherent, conflicts with contextual information or real-world knowledge. The causes of hallucinations are varied, including the use of low-quality datasets and suboptimal data utilizationHuang et al. (2023).

It is widely argued that LLMs primarily acquire knowledge during the pretraining phase, while processes such as supervised fine-tuning and in-context learning are more focused on the extraction and application of existing knowledge, with limited capacity for learning new knowledge. Gudibande et al.Gudibande et al. (2023) found that during supervised fine-tuning, LLMs predominantly learn the stylistic aspects of the fine-tuning dataset rather than factual knowledge. This can lead the model to confidently generate content that is not factually accurate. Research by Kang et al.Kang et al. (2024) suggests that unfamiliar queries can trigger hallucinations in LLMs and that fine-tuning the model on data not covered during pretraining can influence the type of hallucinations produced. Experiments by Yin et al.Yin et al. (2023) and Lee et al.Lee et al. (2023) indicate that using unfamiliar knowledge as a template in in-context learning can degrade the performance of LLMs in such

tasks. Gekhman et al.Gekhman et al. (2024) categorized knowledge into four distinct types based on the model's probability of correctly answering corresponding questions under different decoding strategies. Their experiments unveiled that fine-tuning the model on knowledge that the model has not yet mastered-knowledge that the model cannot correctly answer under a greedy decoding strategy—results in overfitting and notably diminishes the model's accuracy in responding to questions related to mastered knowledge—knowledge that the model can correctly answer under a greedy decoding strategy. Their research indicates that fine-tuning on knowledge that the model is likely to answer correctly enhances test set accuracy, mitigates overfitting, and sustains performance on previously acquired knowledge, as opposed to fine-tuning on all data or a specific type of data. Our methods followed the knowledge classification proposed by Gekhman et al.

Overall, encountering new knowledge after the pretraining phase poses a significant challenge for LLMs, making it difficult both to learn this knowledge and to utilize it effectively during fine-tuning and inference.

## 2.2 QUANTIFYING LLM KNOWLEDGE MASTERY

Assessing the degree to which LLMs have mastered specific knowledge is crucial for both understanding the internal mechanisms of these models and constructing better ones. In scenarios where the ground truth is known, such as in closed-book question-answering tasks, the model's knowledge can be evaluated by generating multiple responses and measuring accuracyKadavath et al. (2022); Gekhman et al. (2024). However, in cases where ground truth is unavailable, such as in real-world applications of LLMs, alternative evaluation methods are required. The confidence score is often used to assess a model's certainty in its specific outputs (such as words, phrases, or sentences) and can serve as an indicator of the model's knowledge mastery. Ideally, a confident score should be well-calibrated, meaning it can accurately reflect the probability that an answer is correct in the real world. Nevertheless, since LLMs are often trained with Reinforcement Learning from Human Feedback (RLHF)Ouyang et al. (2022), they may confidently produce incorrect information, leading to conditional probabilities that are not well-calibrated. Researchers have proposed various methods to address this issue, such as calculating the degree of agreement among different generated outputsLyu et al. (2024), training models to compute confidence scores based on activation statesAzaria & Mitchell (2023), or directly querying the LLM about its confidence in its knowledgeTian et al. (2023). Notably, Kadavath et al.Kadavath et al. (2022) pointed out that the accuracy of repeated responses is a well-calibrated metric for assessing the model's knowledge, which is also the method we adopted to gauge the extent of knowledge mastery.

## 2.3 CONTINUE FINE-TUNING

Our approach involves the issue of continual fine-tuning, where after fine-tuning the model, further fine-tuning is conducted using new data. Continual learning is crucial for models that need to adapt over time, learning from a continuous stream of data without forgetting previously acquired knowledge. However, one of the most significant challenges in this area is the phenomenon known as catastrophic forgetting. This occurs when a model, after being exposed to new data, experiences a significant degradation in performance on tasks it had previously learned and mastered. As the model adjusts its parameters to accommodate new information, it inadvertently overwrites the information and skills it had previously acquired, leading to a loss of accuracy and effectiveness in those earlier tasks. To mitigate this issue, researchers have proposed various methods, broadly categorized into four typesZheng et al. (2024): Replay-BasedShin et al. (2017); Ren et al. (2024), Regularization-BasedMi et al. (2020), Gradient-BasedLee et al. (2021), and Architecture-BasedGeng et al. (2021) approaches. In our experiment, we employed basic experience replay, appropriately lowered the initial learning rate, and set weight decay parameters to prevent overfitting to the new data.

## 3 METHODS

Our experiments are divided into two parts. The first part examines the changes in knowledge types after fine-tuning. After validating our hypotheses in this initial stage, the second part proposes a two-stage fine-tuning strategy based on these findings. Detailed descriptions of our methods are provided in Sections 3.1 and 3.2, respectively. The overall process is illustrated in Figure 1.

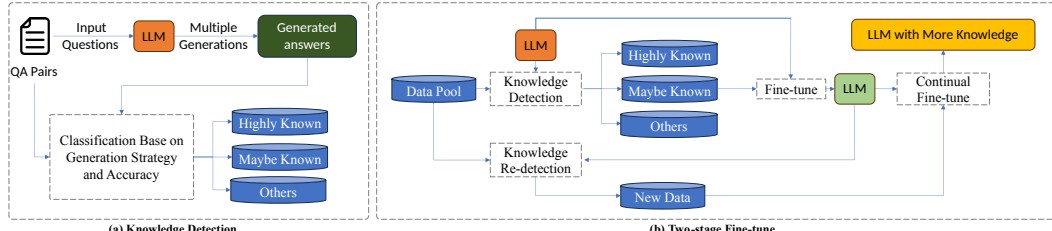

Figure 1: Figure (a) illustrates the method of knowledge classification. For each question-answer pair, multiple prompts were constructed, and the model was queried several times using both greedy and non-greedy decoding methods. Knowledge was then classified based on the accuracy of the responses and the corresponding decoding strategies. Figure (b) depicts the two-stage training process: the first stage involves fine-tuning with knowledge classified as 'Maybe Known', followed by a re-detection of knowledge categories. Based on the re-detection, the training data is augmented, and the model undergoes a second stage of fine-tuning, ultimately resulting in a large language model with a broader mastery of knowledge.

## 3.1 DETECT KNOWLEDGE TYPE CHANGE

Knowledge is interconnected, and it is possible to infer new knowledge from existing information. We propose that during the fine-tuning process, the model may acquire knowledge that it originally did not possess and that is not present in the training dataset for the following reasons:

- Some knowledge not included in the training dataset and initially unknown to the model can be inferred from the knowledge that is present within the training dataset.

- Even if certain knowledge in the training dataset cannot directly infer other knowledge, it can be combined with information acquired by the model during the pretraining phase to infer knowledge that is absent from the training dataset and was originally unknown to the model.

To validate our hypothesis, we first classified the knowledge into four categories—'Highly Known', 'Maybe Known', 'Weakly Known', and 'Unknown'—based on the work of Gekhman et al.Gekhman et al. (2024), with the specific criteria outlined in Table 1. We then fine-tuned the model using data classified as 'Maybe Known'. At the end of each epoch, we evaluated the model's accuracy on the test set. Once the accuracy reached its maximum, we re-evaluated the knowledge types.

## 3.2 TWO-STAGE FINE-TUNING

Based on the experimental results presented in Section 3.1, which will be detailed in Section 4.1, we observed that following fine-tuning, a significant portion of knowledge showed improved mastery, while some knowledge exhibited a decline in mastery. Building on these observations, we proposed a two-stage fine-tuning strategy. In the second stage of fine-tuning, we used data categorized as 'Maybe Known' at the end of the first stage, which includes data originally classified as 'Weakly Known' and 'Unknown' that were excluded from the initial training dataset. We continued fine-tuning with this dataset while implementing a replay strategy with a 0.2 replay ratio for data categorized as 'Highly Known' after fine-tuning. Specifically, at the start of each epoch, we sampled a portion of the 'Highly Known' data according to the replay ratio, mixed it with the rest of the training data, and shuffled it for training. This strategy not only expands the training data but also helps prevent the forgetting of knowledge that the model has already mastered.

## 4 EXPERIMENTS AND ANALYSIS

Sections 4.1 and 4.2 correspond to the experiments and their analyses discussed in Sections 3.1 and 3.2, respectively.

Table 1: Knowledge classification according to Gekhman et al. $(q, a)$ represents a question-answer pair, and $T$ denotes the temperature parameter during decoding. For greedy decoding, $P_{correct}$ represents the probability of the model correctly answering a question $q$. to estimate the probability of correctly answering questions, ten generations were performed. For each generation, prompts were randomly constructed, with each prompt containing four examples. For non-greedy decoding, the temperature was set to 0.5, with a sampling size of 16 and a top-k value of 40. Ten generations were conducted under these conditions as well.

| Type | Defination | Explanation |
| --- | --- | --- |
| Highly Known | $P_{correct}(q, a; T = 0) = 1$ | can be correctly answered using a greedy decoding strategy |
| Maybe Known | $P_{correct}(q, a; T = 0) \in (0, 1)$ | might be correctly answered using a greedy decoding strategy |
| Weakly Known | $P_{correct}(q, a; T = 0) = 0 \land$ $P_{correct}(q, a; T > 0) > 0$ | might be correctly answered by a non-greedy decoding strategy, but not by a greedy one |
| Unknown | $P_{correct}(q, a; T = 0) = 0 \land$ $P_{correct}(q, a; T > 0) = 0$ | cannot be correctly answered even with a non-greedy decoding strategy |

### 4.1 DETECT KNOWLEDGE TYPE CHANGE

#### 4.1.1 EXPERIMENTAL SETUP

At this stage, we broadly followed the experimental methodology of Gekhman et al. We conducted a closed-book question-answering task using the WikiQA datasetYang et al. (2015). The only adjustment we made to the dataset was to train on question-answer pairs with a single answer. During testing, for questions with multiple answers, we recorded a response as correct if the model accurately generated the first answer. For each data point in the training set, we randomly selected four other questions of the same type to create a prompt template and performed multiple generations to assess accuracy. The generation parameters were consistent with those outlined in Table 1. To account for cases where the model generated extra content, we considered an answer correct if the correct response appeared as a substring in the model's output. We used VLLMKwon et al. (2023) to accelerate inference when determining the types of knowledge.

During fine-tuning, we employed LoRAHu et al. (2021) with a rank setting of 64. We used the AdamW optimizer, a cosine scheduler, and set the initial learning rate to 3e-4. All training was conducted on four A100 GPUs, with inference performed on a single A100 GPU, each with 80GB of memory. The batch size was set to 32.

We primarily used the Qwen2 modelYang et al. (2024), a leading open-source model, for our experiments. When the model name is not explicitly mentioned, the model being used is Qwen2-7B. To further validate our conclusions, we also conducted trials using the LLaMA3 modelDubey et al. (2024).

#### 4.1.2 EXPERIMENT RESULTS AND ANALYSIS

The number of each type of knowledge in the training dataset before fine-tuning is shown in Table 2.After fine-tuning for several epochs, we evaluated the model's accuracy on the test set at the end of each epoch. Upon accuracy convergence, we examined the changes in the attributes of each knowledge point. Table 3 presents the experimental results using Qwen2 and LLaMA3.

Analysis of the results presented in the tables reveals the following:

- After several epochs of fine-tuning, a significant portion of knowledge points originally classified as 'Weakly Known' or 'Unknown' were reclassified as 'Maybe Known', with

Table 2: The number of knowledge points for each type before fine-tuning.

| Model | Type | Num | Model | Type | Num |
|---|---|---|---|---|---|
| Qwen2-7B | HighlyKnown | 34426 | LLaMA3-8B | HighlyKnown | 34739 |
| | MaybeKnown | 36897 | | MaybeKnown | 42400 |
| | WeaklyKnown | 25986 | | WeaklyKnown | 24701 |
| | Unknown | 71855 | | Unknown | 67324 |

Table 3: A statistical analysis of the changes in knowledge types after the first stage of fine-tuning. Since distinguishing between 'Weakly Known' and 'Unknown' is costly and less relevant to our approach (as we only need to differentiate between 'Highly Known', 'Maybe Known', or neither when selecting training data), these two categories were grouped together. We have highlighted in bold the parts where the mastery level transitions from a lower level to "Maybe Known."

| Model | Origin Type | New Type | Num |
|---|---|---|---|
| Qwen2-7B | HighlyKnown | HighlyKnown | 27282 |
| | | MaybeKnown | 3952 |
| | | WeaklyKnown & Unknown | 3192 |
| | MaybeKnown | HighlyKnown | 19059 |
| | | MaybeKnown | 9892 |
| | | WeaklyKnown & Unknown | 7946 |
| | WeaklyKnown | HighlyKnown | 3091 |
| | | MaybeKnown | **4889** |
| | | WeaklyKnown & Unknown | 18006 |
| | Unknown | HighlyKnown | 527 |
| | | MaybeKnown | **1932** |
| | | WeaklyKnown & Unknown | 69396 |
| LLaMA3-8B | HighlyKnown | HighlyKnown | 29930 |
| | | MaybeKnown | 4428 |
| | | WeaklyKnown & Unknown | 381 |
| | MaybeKnown | HighlyKnown | 19635 |
| | | MaybeKnown | 19976 |
| | | WeaklyKnown & Unknown | 2789 |
| | WeaklyKnown | HighlyKnown | 1991 |
| | | MaybeKnown | **8607** |
| | | WeaklyKnown & Unknown | 14103 |
| | Unknown | HighlyKnown | 396 |
| | | MaybeKnown | **5165** |
| | | WealyKnown & Unknown | 61763 |

some even transitioning to 'Highly Known'. It is important to note that the model was trained exclusively on data labeled as 'Maybe Known'.

- Some data points initially classified as 'Highly Known' were reclassified as 'Maybe Known', or even to lower categories. This indicates the occurrence of forgetting during the fine-tuning process.

Since we use the frequency of correctly answered questions as a proxy for the model's knowledge mastery, and given that prompt construction during testing is random, with the sampling process being stochastic when setting the temperature parameter, we may observe changes in the classification of some knowledge points even when retesting the same model. To determine how much of these attribute changes are due to training, we retested the knowledge categories using the model without fine-tuning, as shown in Table 4. Although individual knowledge points may change categories between different tests, the number of knowledge points whose categories change remains fairly stable due to the concentration inequality. Therefore, the results of a single retest can indicate the extent to which random factors contribute to changes in knowledge categories. Analysis of the table suggests that a significant portion of the category changes can be attributed to fine-tuning.

Table 4: The differences in results when testing knowledge types twice using the same model.

| Origin Type | New Type | Num |
|---|---|---|
| HighlyKnown | HighlyKnown | 31987 |
| | MaybeKnown | 3190 |
| | WeaklyKnown &Unknown | 0 |
| MaybeKnown | HighlyKnown | 3285 |
| | MaybeKnown | 30006 |
| | WeaklyKnown & Unknown | 5236 |
| WeaklyKnown & Unknown | HighlyKnown | 0 |
| | MaybeKnown | 5285 |
| | WeaklyKnown & Unknown | 97571 |

To investigate the possible reasons behind these changes, we constructed a graph. For question-answer pairs initially classified as 'Maybe Known', as well as those initially classified as 'Weakly Known' but reclassified as 'Maybe Known' after fine-tuning, we represented the entities in both the answers and the questions as nodes. For example, in the question-answer pair "Question: Who performed Rodney Crowell - Greatest Hits? Answer: Rodney Crowell", we extracted 'Rodney Crowell - Greatest Hits' and 'Rodney Crowell' using regular expressions and used them as nodes. For each question-answer pair, we connected the entities in the question and answer with an undirected edge. We categorize the nodes into the following three types:

- **Initial:** Nodes from question-answer pairs initially classified as 'Maybe Known'.
- **Reclassified:** Nodes whose classification changed from 'Weakly Known' to 'Maybe Known'.
- **Linked Reclassified:** Nodes reclassified from 'Weakly Known' to 'Maybe Known' and connected to 'Initial' nodes.

The Table 5 shows that after several epochs of training, most nodes that transitioned from 'Weakly Known' to 'Maybe Known' are closely connected to nodes originally classified as 'Maybe Known', which aligns with our hypothesis.

Table 5: The connections between knowledge initially classified as 'Maybe Known' and the knowledge that transitioned from 'Weakly Known' to 'Maybe Known' after the first stage of fine-tuning.

| Label Type | Initial | Reclassified | Linked Reclassified |
|---|---|---|---|
| **Count** | 43,005 | 5,121 | 3,968 |

## 4.2 TWO-STAGE FINE-TUNING

### 4.2.1 EXPERIMENTAL SETUP

During continual fine-tuning, we reduced the initial learning rate to 15e-5 and set the weight decay to 0.01. We set the replay ratio to 0.2 and continued using LoRA with a rank of 64. To minimize the influence of random factors when evaluating the model's accuracy, we generated a fixed prompt test set at the beginning of the experiment using random seed 42.

### 4.2.2 EXPERIMENTS AND ANALYSIS

We first validated the limitations of one-stage fine-tuning, specifically when using only data initially classified as 'MaybeKnown', as also recommended by Gekhman et al., which outperforms training on all data or other types of data. Our results, shown in Table 6, indicate that increasing the number of fine-tuning epochs does not improve model performance. It can be observed that after 10 epochs (in fact, the highest test accuracy is achieved by the end of the eighth epoch), increasing the number of training epochs does not enhance accuracy when using an early stopping strategy. Moreover,

without early stopping, extending the training epochs leads to overfitting on the training set, resulting in a decline in test accuracy.

Table 6: The peak accuracy and final accuracy at the end of training across different epochs during fine-tuning on data classified as 'Maybe Known'.

| Training Epoch | Max Acc | Final Acc |
|---|---|---|
| 10 | 32.98 | **32.74** |
| 15 | 32.99 | 31.21 |
| 20 | **33.00** | 30.73 |
| 25 | 32.99 | 30.50 |

Table 7: Accuracy results obtained using two-stage fine-tuning, one-stage fine-tuning, and without fine-tuning.

| Model | Stage | Max Accuarcy |
|---|---|---|
| Qwen2 | Origin | 29.93 |
| | One-Stage | 32.98 |
| | Two-Stage | **33.80** |
| LLaMA3 | Origin | 31.55 |
| | One-Stage | 39.16 |
| | Two-Stage | **39.59** |

We then conducted the two-stage fine-tuning experiment, collecting the maximum accuracy achieved after the second stage (typically occurring at the end of the first to third epochs), and compared it with the results from the one-stage fine-tuning. The results are presented in Table 7. The experimental results demonstrate that our method significantly improves the model's test accuracy.

Since the data used for two-stage fine-tuning includes knowledge classified as 'Weakly Known' after the first stage as well as knowledge classified as 'Highly Known', we aimed to determine whether the improvement in model accuracy after the two-stage fine-tuning is due to the acquisition of new knowledge or the reduction of forgetting through data replay of previously mastered knowledge. To investigate this, we designed various two-stage fine-tuning strategies and conducted experiments. The different strategies are outlined below, and the experimental results are presented in Table 8.

- **Strategy 1:** Fine-tuning with data that remained classified as 'Maybe Known' after fine-tuning and was initially categorized as 'Highly Known', 'Maybe Known', or 'Weakly Known'.

- **Strategy 2:** Fine-tuning with data that remained classified as 'Maybe Known' after fine-tuning and was initially categorized as 'Highly Known' or 'Maybe Known'.

- **Strategy 3:** Fine-tuning with data that remained classified as 'Maybe Known' after fine-tuning and was initially categorized as 'Maybe Known' or 'Weakly Known'.

- **Strategy 4:** Fine-tuning with data that remained classified as 'Maybe Known' after fine-tuning.

- **Strategy 5:** Fine-tuning with data that remained classified as 'Maybe Known' after fine-tuning, while replaying 20% of data classified as 'Highly Known' after fine-tuning. Strategy 5 is the one used in the previous experiments.

Table 8: The maximum accuracy achieved after second-stage fine-tuning using datasets constructed with different strategies.

| Strategy | One-Stage | Strategy 1 | Strategy 2 | Strategy 3 | Strategy 4 | Strategy 5 |
|---|---|---|---|---|---|---|
| **Max Acc** | 32.98 | 33.62 | 33.24 | 33.29 | 33.62 | **33.80** |

To explore the impact of two-stage fine-tuning on the model's accuracy concerning knowledge it originally mastered, we classified the data in the test set using the un-tuned model and then compared the model's test accuracy on different types of knowledge after applying different strategies for two-stage fine-tuning. The results are shown in Table 9.

Based on the results presented in the Table 8 and Table 9, we can conclude the following:

- Whether by using the forgotten data after fine-tuning as additional data for continued fine-tuning, employing newly improved data as supplementary data, or combining both approaches, it is possible to effectively enhance the model's performance. The improvement

Table 9: The model's accuracy on test data across different categories, which were determined using the un-finetuned model. It can be observed that all strategies, except for Strategy 2, improve the model's accuracy across all categories of test data during the second stage of fine-tuning. Strategy 2 did not utilize data that was initially classified as 'Weakly Known' but showed improved mastery.

| Stage | HighlyKnown | MaybeKnown | WeaklyKnown | Unknown |
|-------|-------------|------------|-------------|---------|
| One-Stage | 85.03 | 56.90 | 18.38 | 1.62 |
| Strategy 1 | 86.76 | 56.94 | 19.32 | 1.95 |
| Strategy 2 | 86.96 | 56.77 | 17.86 | 1.59 |
| Strategy 3 | 85.67 | 57.06 | 18.88 | 1.77 |
| Strategy 4 | 86.59 | 56.35 | 19.90 | 2.13 |
| Strategy 5 | 86.94 | 57.00 | 19.64 | 2.15 |

in model performance after two-stage fine-tuning results from both the acquisition of new knowledge and the mitigation of forgetting existing knowledge during the fine-tuning process.

- Combining both approaches—augmenting the training data with knowledge that shows improved mastery after the first stage of fine-tuning, while simultaneously replaying fully mastered data to mitigate forgetting—proves to be the most effective strategy.

- Regardless of whether data initially classified as 'HighlyKnown' is used during the second stage of fine-tuning, the model's accuracy on test data initially classified as 'HighlyKnown' is not diminished—in fact, it is improved.

While testing the model's accuracy on the test set can effectively reflect the model's capabilities, it does not entirely align with our objective. Our aim is to expand the amount of data that LLMs can utilize during supervised fine-tuning, alleviate the negative impact of the model's weaker learning ability during the fine-tuning stage, and thereby enhance the potential of LLMs from a data perspective. Hence, in addition to the accuracy on the test set, we also need to understand the model's mastery of knowledge in the training set. Therefore, we also tracked the number of knowledge points of each type in the training set at different stages, as presented in Table 10. The findings suggest that compared to one-stage fine-tuning, two-stage fine-tuning can:

- significantly increase the number of knowledge classified as 'Highly Known' (approximately **7.5%**.If solely considering incremental enhancements, it reaches approximately **24%**.), representing question type that the model is highly likely to answer correctly.This indicates that the method significantly enhances the knowledge mastered by the model.

- reduces the proportion of data classified as 'Weakly Known' and 'Unknown'. This implies an expansion of the range of data that can be used for fine-tuning.

Table 10: The number of different types of knowledge at various stages.From the table, it can be observed that after two-stage fine-tuning, the classification of 'Highly Known' has significantly increased, while the other types have decreased.

| Type | Origin | One-Stage | Two-Stage |
|------|--------|-----------|-----------|
| HighlyKnown | 34426 | 49959 | **53691** |
| MaybeKnown | 36897 | 20665 | **18288** |
| WeaklyKnown & Unknown | 97841 | 98540 | **97185** |

### 4.2.3 MULTIPLE-ROUNDS FINE-TUNING ATTEMPTS

Building on the most effective Strategy 5, we attempted multiple rounds of fine-tuning. After completing the two-stage fine-tuning process, we again assessed the changes in knowledge categories and iteratively applied the fine-tuning method described earlier, aiming to maximize the model's performance. However, the results indicated that multiple rounds of fine-tuning did not lead to further improvements in model performance. The results are shown in Table 11.

Table 11: Accuracy results for one-stage, two-stage, and multi-round fine-tuning.

|  | One-Stage | Two-Stage | Multi-Rounds |
|---|---|---|---|
| **Accuracy** | 32.98 | **33.80** | 33.79 |

We hypothesize that this phenomenon may be due to the fact that, after the first round of fine-tuning, the model has already improved its mastery of knowledge as much as possible given its current capabilities and the content of the dataset. After the second round of fine-tuning, there was little further improvement in knowledge mastery. The lack of new data for continued fine-tuning led to a scarcity of fresh information, and further training on the existing data resulted in overfitting. We analyzed the changes in knowledge types in Table 12. As seen in Table 12, compared to the first stage, the number of knowledge types that changed before and after the second stage of fine-tuning significantly decreased, supporting our hypothesis. This rapid convergence also highlights the efficiency of our method.

Table 12: The changes in knowledge types before and after the second stage of fine-tuning.

| Origin Type | New Type | Num |
|---|---|---|
| HighlyKnown | HighlyKnown | 47335 |
|  | MaybeKnown | 2527 |
|  | WeaklyKnown &Unknown | 97 |
| MaybeKnown | HighlyKnown | 6131 |
|  | MaybeKnown | 13161 |
|  | WeaklyKnown & Unknown | 1373 |
| WeaklyKnown & Unknown | HighlyKnown | 225 |
|  | MaybeKnown | 2600 |
|  | WeaklyKnown & Unknown | 95715 |

## 5 DISCUSSION

We proposed that fine-tuning can alter the classification of knowledge that was not directly involved in the fine-tuning process, and we conducted experiments to validate this hypothesis. Based on this insight, we refined the fine-tuning process, which led to improvements in both model accuracy on the test set and the model's mastery of the knowledge in the fine-tuned dataset. However, our experiments remain largely qualitative, focusing primarily on demonstrating the feasibility of this approach.

Compared to the specific domain datasets that might be used for fine-tuning large language models in real-world applications, the WikiQA dataset we used is relatively loosely structured. This dataset consists of simple question-answer pairs with a limited set of question formats, and the questions lack the strong internal coherence typically found in domain-specific datasets. In such specialized datasets, data is often more focused within a particular domain, where the tighter connections between knowledge might lead to more significant changes in knowledge classification during fine-tuning.

Moreover, for knowledge that falls outside the scope of closed-book question-answering tasks, determining whether the knowledge is present in the model is more challenging. These differences indicate that our approach has broader potential applications but also faces new challenges.

Furthermore, when conducting continual learning, effectively addressing catastrophic forgetting requires a balanced approach between new and old data. Better application of continual fine-tuning methods is necessary to achieve this balance. In our experiments, we only utilized basic data replay and moderately reduced the learning rate, indicating that there is still significant room for improvement.

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
