# OpenReview forum: "Gradual Learning: Optimizing Fine-Tuning with Partially Mastered Knowledge in Large Language Models"
_ICLR.cc/2025/Conference — ICLR 2025 Conference Withdrawn Submission_

### Official Review · Reviewer_n8zH · 2024-10-18

**Soundness:** 1
**Presentation:** 1
**Contribution:** 1
**Rating:** 1
**Confidence:** 5

**Summary:**

This paper first designed an experiment to validate their hypothesis that as the model’s capacity to utilize existing knowledge improves during fine-tuning, previously unmastered knowledge may become more understandable. Then they propose a two-stage fine-tuning method to address this.

**Strengths:**

* knowledge probing/ augmentation is an important research problem for large language models.

**Weaknesses:**

* The presentation in this paper is very poor. Actually, it's very hard for me to follow them due to the poor logic in the abstract and introduction. Also, figures and tables are not well-designed.

* Many format problems. There is no space between text and citation; cited and citep are not properly used in this paper; Some citations are even raw text (line 242)

* The proposed method and findings are trivial. And the experiment settings that tune the model in the test set are also problematic to me.

**Questions:**

I suggest the authors submit this paper maybe tot a workshop.

---

### Official Review · Reviewer_9bFL · 2024-10-30

**Soundness:** 1
**Presentation:** 2
**Contribution:** 1
**Rating:** 3
**Confidence:** 4

**Summary:**

The paper investigates the challenges of finetuning in language models, noting that introducing new knowledge can lead to hallucinations and degraded performance. The authors hypothesize that reinforcing partially mastered knowledge could improve the model’s performance with new knowledge. They use a two-phase approach: (1) Knowledge detection (types) and (2) Two-stage finetuning with 'MaybeKnown' and 'HighlyKnown' Knowledge Replay. Primarily using Qwen-2, they analyze the contributions of each finetuning stage.

Writing and Clarity:

  - The paper suffers from unclear writing, especially in the abstract and motivation sections.
  - Poor organization and coherence in presenting experiment motivation, methods, and results, impacting the overall flow.
  - Findings are not clear and are not connected to the observations enough.


Reliance on Previous Contribution:

    - Misinterpret ‘MaybeKnown’ recommendation (line 374) for improving performance on less known knowledge -  Gekhman et al 2024 claimed that the ‘MaybeKnown’ helps to less hallucinate data the model already knows.

Explanation and Motivation:

 Lacks clear motivation for choosing specific strategies:

    - No justification for preferring a replay-based approach for catastrophic forgetting (line 154).
    - Insufficient explanation for focusing on the ‘MaybeKnown’ category over ‘HighlyKnown’ for finetuning.
    - Weak or missing rationale for strategies 1-5 (lines 403-416).
    - Justification is absent for selecting the first answer in multi-answer QA (lines 245-246).
   - Foundational intuition (Section 3.1) for acquiring new knowledge from partial knowledge is weak and lacks references.


Weak Results:
  - The proposed method shows modest improvements (~4-7%) in accuracy (Table 7) which is not strongly associated with a specific factor.
  - Results lack consistency: replay strategies sometimes decrease performance (e.g., Table 9, Strategies 4 and 5 for WeaklyKnown).
  - Small performance changes in Table 8 might result from various, unexamined factors.
  - No analysis of performance differences between models (Qwen and LLaMA).
- Not robust - used only one dataset (closed).


Results and Clarity:
  - Results are presented in raw counts (Tables 3, 4, 5, 10), without proportions, making the transition and interpretation difficult.
  - Graph construction (line 338) is unclear and does not clearly support the hypothesis; the contribution of node reclassification is ambiguous.

**Strengths:**

- Proposes Interesting intuition: integrating partially-known knowledge can boost performance on new known knowledge
        - Presents a clear, structured methodology in Fig. 1

**Weaknesses:**

The paper's contribution appears limited, as the concept of using ‘MaybeKnown’ knowledge to enhance test-set performance has already been established by Gekhman (2024). The newly proposed two-stage finetuning with a replay method results in only minor improvements (~1-2%), which, coupled with weak, non-robust analysis and disorganized writing, supports a recommendation for rejection.

**Questions:**

Formatting:

Missing spaces and inconsistent citation format throughout the paper (e.g., Intro lines 40, 45, 51).

Terminology:

Define terms before using them:
“KG” is used without explanation (line 67).
Knowledge types (line 75).
Prompt template (line 247).
The second stage of finetuning in the Introduction is unclear and would benefit from clarification.

Methodology:

- Method Section 3.2: clarify data categories used in the finetuning steps (lines 203-206).
- Confirm if knowledge re-detection after each finetuning stage is performed on the training set—currently unclear.
- Definitions and Supplementary Information:
   Table 1 is identical to the one in Gekhman’s work (Figure 2(a)), and presented in the main paper which might be mistakenly interpreted as
   some contribution.
- Define “Origin” in Table 7. Does it represent baseline performance without finetuning on ‘MaybeKnown’?
- Consider adding a supplementary section, e.g., for explaining prompt template creation (line 247).

Results Interpretation:

- Table 8: Strategy 1 and Strategy 4 have identical scores—does this imply an effect specific to ‘HighlyKnown’ knowledge?
Divide the main results from analyses.

---

### Official Review · Reviewer_KNQC · 2024-11-03

**Soundness:** 3
**Presentation:** 2
**Contribution:** 2
**Rating:** 3
**Confidence:** 3

**Summary:**

This paper tackles the challenges that LLMs primarily acquire knowledge during pretraining and often struggle to integrate new information effectively in later stages such as fine-tuning and inference times. This can lead to issues like hallucinations and decreased performance (catastrophic forgetting).
The core argument of this paper is that fine-tuning LLMs on partially mastered knowledge can help them leverage their existing knowledge base and reasoning capabilities to comprehend new concepts. The authors propose a two-stage fine-tuning to test this hypothesis.
Stage 1: The model is fine-tuned on data representing knowledge it partially understands, meaning it has some chance of answering related questions correctly without fine-tuning.
Stage 2: The training data is augmented with data that demonstrates improved mastery after the first stage, including knowledge initially classified as less well-understood. This augmented dataset is then used for a second round of fine-tuning.
The results on the WikiQA dataset show that the two-stage fine-tuning method leads to improved test accuracy, enhanced knowledge mastery and mitigation of forgetting.

**Strengths:**

The motivation of this paper is well articulated, and it is important indeed such as how to avoid hallucination and catastrophic forgetting when LLMs are fine-tuned after pre-training.

**Weaknesses:**

1. Limited generalizability due to focus on a single closed-book question answering: The experiments rely heavily on the WikiQA dataset, which is specifically designed for closed-book question answering. This focus raises concerns about the generalizability of the results to other knowledge domains and tasks.

2. Tiny Incremental Improvements: While the two-stage fine-tuning method shows improvements in test accuracy and knowledge mastery compared to one-stage fine-tuning, these improvements are relatively small. For example, Table 7 in the source shows that for the Qwen2 model, the accuracy improvement from one-stage to two-stage is smaller than 1%. This raises the question of whether the added complexity and computational cost of the second stage are justified by such marginal gains.

3. Practicality of the method: Given the need for pre-classification, the practicality of the two-stage method for tasks that are not already determined is debatable.

**Questions:**

- The compiled \cite looks broken throughout the paper.

---

### Official Review · Reviewer_CPVv · 2024-11-05

**Soundness:** 3
**Presentation:** 2
**Contribution:** 2
**Rating:** 5
**Confidence:** 3

**Summary:**

This paper attempts to address the knowledge forgetting problem in the scenario of continual fine-tuning, where we might lost some information/performance if we do continual fine-tuning.

They perform 2-stage fine-tuning:
1. Categorize the data into certain categories to understand how the model knows about the previous knowledge.
2. The second stage with mix up with some replay data to continue fine-tuning.

**Strengths:**

1. Efforts in designing strategies for how we fine-tuning with the replay data and the training data. We can see that, some unknown data change to MaybeKnown, justifying the  hypothesis.
2. Retain the model original performance, and also comparable performance with SFT over new data (1-stage fine-tuning).

**Weaknesses:**

1. Paper is not well written, especially the citation position.
2. It’s not practical to assume we have the original replay data. For example, if we use a pre-trained model (Qwen), they will not release the SFT data for the community. if we want to improve the model performance on QA using our new data, we probably don’t have original SFT data for replay. The assumption seems not practical.

**Questions:**

Maybe the author can help answer the second weakness.

---

### Official Review · Reviewer_W8Wn · 2024-11-05

**Soundness:** 2
**Presentation:** 1
**Contribution:** 1
**Rating:** 3
**Confidence:** 4

**Summary:**

The paper proposes a two stage finetuning strategy which supposedly improves the models knowledge retention capacity. Using the taxonomies introduced by [1], the paper conducts multi-stage finetuning ablations on the different sequence of subsets.

[1] https://arxiv.org/pdf/2405.05904

**Strengths:**

- Conducts several ablations on sequential multi-stage finetuning on easy to hard dataset subsets.

**Weaknesses:**

- The contributions of the paper is minimal, given they performed ablations on data taxonomies introduced by [1]. At best, the work is an extension of the ablations performed by [1] themselves (see Section 5 of the paper).
- Only one downstream dataset (WikiQA) is considered, which limits the broader applicability of the approach.
- The paper is very poorly written, with large swaths of the text borderline unintelligible. Specially sections 3.1, 3.2, 4.1.2 need a lot of work to make the flow understadable.
- Numerous evidence on poor quality of writing / organization: Table on page 5 is redundant, statistical analysis on changes from one subset to the next is not highlighted well in Tables 2,3,4,12. It is highly unclear what I'm looking at - the change should be presented _relatively_, not as absolute numbers which is meaningless.



[1] https://arxiv.org/pdf/2405.05904

**Questions:**

- In writing, please add a space between all citations - it becomes hard to read the full sentences!
- What is the overall recommendation in terms of finetuning for multiple stages? It is not clear from the text.

---

### Note · Authors · 2024-12-09

I have read and agree with the venue's withdrawal policy on behalf of myself and my co-authors.